# The Effects of Providing Advance Notice and Stress-Coping Traits on Physiological Stress of Patients during Dental Treatment

**DOI:** 10.3390/ijerph19052540

**Published:** 2022-02-22

**Authors:** Sachi Umemori, Kanako Noritake, Ken-ichi Tonami, Son Hoang Le, Masayo Sunaga, Yasuyuki Kimura, Yuna Kanamori, Ayako Sekiguchi, Hiroshi Nitta

**Affiliations:** 1Oral Diagnosis and General Dentistry, Tokyo Medical and Dental University Hospital, Tokyo 1138549, Japan; sachi.u.gend@tmd.ac.jp (S.U.); noritake.irm@tmd.ac.jp (K.N.); kimuperi@tmd.ac.jp (Y.K.); kanamori.ope@tmd.ac.jp (Y.K.); 2Department of Oral Surgery, Faculty of Odonto-Stomatology, University of Medicine and Pharmacy at Ho Chi Minh City, Ho Chi Minh City 72714, Vietnam; lehoangson@ump.edu.vn; 3Department of Educational Media Development, Graduate School of Medical and Dental Sciences, Tokyo Medical and Dental University, Tokyo 1138510, Japan; sunaga.emdv@tmd.ac.jp; 4Department of Oral Diagnosis and General Dentistry, Graduate School of Medical and Dental Science, Tokyo Medical and Dental University, Tokyo 1138549, Japan; a.sekiguchi.gend@tmd.ac.jp

**Keywords:** stress, R-R interval on ECG, psychological test

## Abstract

Patients tend to feel stress in association with dental treatment due to uneasiness and fear. We investigated the effects of providing advance notice and stress-coping traits on the physiological stress of patients during dental treatment. Sixty non-dental professionals (male, *n* = 26; female, *n* = 34; mean age, 49.9 years) were recruited for this study and informed consent was obtained. Subjects were given simulated dental treatment including three stimuli, air, percussion and running of an air turbine, with or without advance notice of the stimulation during dental treatment. Real-time sympathetic nerve activity (SN) and parasympathetic nerve activity (PN) during the treatment were measured using a biological information monitor. The stress-coping traits of each subject were examined using the Lazarus-Type Stress Coping Inventory (SCI). Correlations between the nerve-activity and scores of eight stress-coping strategies of SCI and the presence/absence of advance notice were analyzed. Age, types of stimuli and order of stimuli significantly affected SN, while age, types of stimuli, and the pattern of stimulation significantly affected PN. The interaction of the stress-coping trait and presence/absence of advance notice significantly affected PN. Providing advance notice may have different effects on physiological stress depending on how the patient copes with stress.

## 1. Introduction

In dental treatment, patients tend to feel fear and anxiety because surgical treatment is often performed in a dental clinic while the patient is conscious. At times, huge and unreasonable fear or anxiety during treatment can become an excessive stressor and induce symptoms of dental phobia that disturbs routine dental checkups [1,2,3,4,5,6]. One report pointed out the similarity of dental anxiety to post-traumatic stress disorder (PTSD) [7]. Thus, reducing patient stress during dental treatment is desirable for realizing safe and peaceful medical care and improving the patients’ QOL. It has been known that providing patients with information about the procedure is a useful way to reduce patients’ stress during dental treatment because of an increasing sense of predictability [3,8]. Lazarus states in his transactional model of stress and coping that the stress reaction is not determined only by the stressor itself, but also by cognitive appraisal and coping. That is, the effect of advance notice on stress reduction may depend on patients’ characteristics. As a tool to survey such psychological traits, Lazarus introduced the Lazarus-Type Stress Coping Inventory (SCI), which enables analysis of individuals’ stress coping styles. However, to the best of our knowledge, only a few studies have reported the effects of advance notice [9] and patients’ stress-coping traits on patients’ stress reactions [10].

Stress is often classified into two types: psychological stress and physiological stress [11,12]. Many trials have been undertaken to quantify stress and the uneasiness of patients using various psychological stress indexes. It should be recognized, however, that many of the psychological stress indexes used had self-reported measures and a limited focus, mainly because of their subjectivity [13]. Therefore, there is a need for an index that can quantify patient stress objectively. As objective measures of internal stress, physiological stress indexes, such as electrodermal activity (EDA) [14,15], salivary stress markers (e.g., cortisol [16] and α-amylase [17]), electromyography, and electroencephalography (EEG) [18,19,20] have been utilized. Using such physiological stress indexes, we studied the real-time monitoring of physiological stress to reduce patient stress during dental treatment. As one of the achievements, Ozaki reported the relationship between physiological stress and psychological stress, that is, how the psychological stress coping traits of the patients influence the effect of preceding information on the sympathetic nerve activities measured by EDA [15]. On the other hand, some disadvantages of EDA became clear through the experiment. That is, measurements of EDA are easily affected by the condition of the skin to which the electrode is attached (e.g., moisture or thickness of the horny layer). Such sensitivity to skin conditions means that time is required to calibrate the analysis to each individual.

Recently, the usefulness of a frequency analysis of the R-R (FAR) interval on electrocardiography (ECG) has been reported in physiological stress assessment [21,22]. The advantage of FAR is that it noninvasively and conveniently enables real-time stress monitoring of patients during dental treatment. Sekiya et al. reported the usefulness of the method for the continuous assessment of subjective discomfort levels that could not be detected by EEG or corrugator muscle electromyography [20]. Another advantage of FAR is that it can separately measure real-time sympathetic and parasympathetic nerve activity during dental treatment. Although FAR has been widely used in psycho-physiological research due to its convenience [23,24], the use of FAR in dentistry has still been limited. Therefore, in this study, the effects of providing advance notice and SCI on the physiological stress of patients during dental treatment were investigated using this biological information monitor. We hypothesize that the effects of this advance notice on patients’ physiological stress will differ depending on their stress-coping traits.

## 2. Materials and Methods

### 2.1. Subjects

The subjects were recruited from those who visited our university to have general treatment and maintenance. One of the two researchers obtained informed consent from the participants before the examination. The subjects were 60 non-dental professionals (male, *n* = 26; female, *n* = 34; mean age, 49.9 years, ranging from 23 to 79 years old). We excepted those with dental phobia who had strong dental treatment anxiety which prevented them from going to the dental clinic.

### 2.2. Patterns of Stimulation as Simulated Dental Treatment

Three types of stimuli that simulate a dental examination or treatment were predetermined as follows: blowing air to the right side of the mandibular first molar for 5 s (Air), percussion on the right side of the mandibular first premolar with the handle of tweezers four times (Per), and no-load running of an air turbine beside the right side of the mandibular first molar with a water spray for 10 s (Tur). For each stimulus, two conditions were set. One was with advance notice, and the other was without advance notice. The contents of advance notice of “Air”, “Per”, and “Tur” were, “I am going to blow air on your tooth”, “I am going to tap your tooth”, and “I am going to use a drill”, respectively. The advance notice was set to be given to the subjects 3 s before each stimulus. The interval between each stimulus was 30 s.

Three patterns of stimulation were prepared. In each of them, the subjects were to be exposed to stimuli 12 times. For each simulated pattern, stimuli were randomly selected from the combination of three types of stimuli (Air, Per, Tur) and two conditions (presence/absence of advance notice) (Table 1).

### 2.3. Frequency Analysis of R-R Interval on ECG (FAR) under Simulated Dental Treatment

One of the three simulated patterns was randomly assigned to each subject. Before the simulated dental treatment, the subjects lying on the dental chair in a supine position wore an eye mask. During simulated dental treatment, stimuli were applied according to the assigned simulation pattern with monitoring of ECG using a biological information monitor (Relax Meijin, Croswell Co. Ltd., Tokyo, Japan). At the same time, the real-time-FAR of the subject was conducted and the amount of change in low frequency (LF: 0.04–0.15 Hz) and high frequency (HF: >0.15 Hz) were continuously recorded. In the subsequent analysis, LF/HF was calculated and standardized, then used as an index of sympathetic nerve activity (SN). The standardized HF was used as an index of parasympathetic nerve activity (PN) [25,26].

### 2.4. Investigation of Stress-Coping Traits

The stress-coping traits of each subject were examined using the Lazarus-Type Stress Coping Inventory (SCI), the Japanese version of the Ways of Coping Questionnaire as a psychological stress index [27] before simulated dental treatment. The SCI is a questionnaire that evaluates ways of coping. It consists of 64 questions asking about behavior that the subjects remember exhibiting in situations in the past when they have felt stress. The examples of questions include the following: “I made a meticulous plan to solve the problem“/”I tried to forget about the problem”. The subjects were asked to answer as a rating on a 3-point scale consisting of ‘yes’, ‘slightly yes’, and ‘no’. The answers of SCI were calculated and the scores of eight strategies for stress coping were obtained, respectively. The eight strategies were as follows: Planned problem-solving (Pla: deliberate analytic efforts to alter or remedy the situation), confrontive coping (Con: efforts to identify the cause of the problem and eliminate the source), seeking social support (See: efforts to seek informational, tangible, and emotional support from others), accepting responsibility (Acc: acknowledging one’s role in the problem with rectification), self-controlling (Sel: efforts to control one’s feelings and actions), escape-avoidance (Esc: wishful thinking to escape or avoid a problem), distancing (Dis: efforts to detach oneself and to minimize the significance of the situation), and positive reappraisal (Pos: efforts to create a positive meaning and personal growth). A higher score suggests that the respondent has stronger stress coping strategy traits.

### 2.5. Statistical Analysis

The SN and PN activities corresponding to each stimulus were defined as the absolute value of change of SN and PN during the period in which the dental instrument entered the oral cavity, applied stimulation, and was set back in the primary position. For the investigation of factors that influence SN and PN, each of 12 stimuli in all subjects were statistically analyzed as follows: A generalized linear mixed-effect modeling approach (GLMM) was used to deal with any clustering effect (subjects) (*p* < 0.05). Stimuli were set as level 1 and subjects were set as level 2. The first-order autoregressive structure was employed as covariance structures of residuals within the subject level. First, correlations between the nerve-activity indices (SN and PN) and each factor (sex, age, type of stimulus, presence/absence of advance notice, the pattern of stimulation and the scores of eight strategies for stress coping) were analyzed separately. Next, a GLMM was done including all the factors to reveal their associations with each SN and PN. All analyses were performed with SPSS 24 (Japan IBM Co., Tokyo, Japan).

## 3. Results

### 3.1. The Results of the Univariate Analysis

The mean values of SN and PN for each item are shown in Table 2. SN showed significant differences according to sex, while PN showed significant differences according to sex and the pattern of stimulation (*p* < 0.05). Among the SCI traits, none of them were positively correlated with SN and PN.

### 3.2. The Results of the Multivariate Analysis

Table 3 shows the results of GLMM regarding SN as an objective variable including all the factors simultaneously. The age, type of stimulus, and order of stimuli significantly affected SN, while sex, advance notice, pattern of stimulation, stress-coping traits, and the interaction between stress-coping traits and presence/absence of advance notice did not affect SN. Table 4 shows the results of the GLMM analysis with PN as an objective variable. The age, type of stimulus, and pattern of stimulation significantly affected PN, while sex, advance notice, order of stimuli, and stress-coping traits did not affect PN. The interaction of stress-coping traits, Pos, and presence/absence of advance notice showed a significant effect on PN.

## 4. Discussion

In this study, advance notice did not demonstrate a significant effect on SN and PN with univariate analysis, but with the multivariate, the interaction term of Pos among SCI traits and advance notice showed a significant positive correlation with PN. Therefore, our hypothesis is supported. Leahy and Woodruff described that identifying the stressor is the first step of cognitive therapy for stress. By identifying stressors using verbalization, amorphous anxiety, or fear in one’s mind can become more focused, which helps the person think more rationally [28]. In the present study, advance notice assisted subjects in identifying the stressor. The next step of cognitive therapy for stress is stress appraisal and coping, which was firstly advocated by Lazarus and Folkman [29,30]. According to their theory, individual responses to identified stressors are processed through two cognitive appraisals. The primary appraisal is for potential harm or threat of stressors, and the secondary appraisal is for options available to cope with a stressor. The stress coping trait, Pos, is associated with positive reappraisal. It was also reported that Pos is related to a view of life that both development and progression come after overcoming difficulties. Thus, it was suggested that subjects with a higher Pos score become relaxed by the positive interpretation of provided information, which results in increasing PN.

In a previous report, the authors revealed that FAR during mandibular third molar surgery did not show a significant relationship with the amount of preoperatively provided information concerning the surgical procedure and its risks [9]. The result that information-giving did not affect FAR could possibly have been due to not considering subjects’ stress-coping traits. The timing of information-giving also affects patients’ stress responses, which might have been another reason for the discrepancy from the result of the present study [3]. In the present study, age positively correlated with SN and negatively with that of PN; this agrees with the findings of a previous study [9]. On the other hand, it has been reported that both SN and PN decline with age in normal healthy persons [31,32]. There is a possibility that FAR trends by age can differ under stressful conditions. Both SN and PN did not change significantly with sex, which was incongruent with a previous report which indicated that females have higher levels of parasympathetic nerve activity and lower levels of sympathetic nerve activity [9]. The fact that the stressors used in this study were simulated and non-invasive might have been a considerable reason for the discrepancy.

The type, order, and pattern of stimuli which significantly affected both SN and PN had clearer effects than advance notice in this study. Therefore, it could be considered that providing information alone is not sufficient to reduce a patient’s stress. We may need to perform further studies with masking of stress stimuli. Improving how the stimuli are applied could be more effective than being careful about how the information is given to achieve more patient-friendly dental practice.

Heart rate variability is said to arise from interference of the respiratory system signals under the control of autonomic cardiovascular systems. FAR can separate the performance of both systems from each other. That is, the obtained low-frequency component represents control of the antagonism of the sympathetic and parasympathetic nerves produced by baroreceptors at the aortic arch, while the high-frequency component represents the parasympathetic nerve activity caused by the pulmonary receptor. To reduce patient stress, we investigated the effects of providing advance notice and stress-coping traits on the physiological stress of patients during dental treatment using a biological information monitor that can separately measure real-time sympathetic and parasympathetic nerve activity as indices of physiological stress. However, it must be noted that the obtained SN is a relative value that is affected by PN. This means that LF/HF, which is used as an index of SN, increases not only with hyperactivity of the sympathetic nervous system, but also with the suppression of PN [33]. The obtained results suggested that the effects of advance notice on stress reduction potentially depend on the subject’s stress-coping traits. Therefore, when providing advance notice during dental treatment, the provision of individualized information that is suited to personal stress-coping traits—as opposed to uniform information—is necessary for stress reduction.

This study has the following limitations. First, only painless stressors were used, and local anesthesia was not included in this simulated treatment activity, because epinephrine contained in local anesthesia vail could make a substantial contribution to vital signs during dental treatment. On the other hand, administering local anesthesia by intra-oral injection is one of the most stressful aspects of the dental treatment experience [34]. In the future, it is necessary to devise a study that can accurately analyze stress under local anesthesia. Next, the subjects in this study were recruited from those who would not avoid dental treatment due to fear; however, no prior oral examination was conducted. Since people with dental treatment phobia have a statistically significantly higher number of cavities and missing teeth [35], further examination is needed to reveal effect of dental history.

Under the limitations of this study, we showed that the effects of advance notice of treatment just before treatment on physiological stress will depend on the stress-coping traits of individual patients. Further investigations using physiological stress measurement in a real-world clinical setting are needed to make dental clinical treatment more patient-friendly.

## 5. Conclusions

The sequential provision of information about the procedure may have different effects on physiological stress depending on a patient’s stress-coping style. The attributes of the stimuli had a greater effect on the physiological stress of the subject than advance notice.

## Figures and Tables

**Table 1 ijerph-19-02540-t001:** Patterns of stimulation used for simulated dental treatment.

Order of Stimuli		1	2	3	4	5	6	7	8	9	10	11	12
pattern 1	types of stimuli	Per	Tur	Per	Air	Tur	Air	Tur	Air	Per	Tur	Air	Per
	presence/absence of advance notice	−	+	+	−	−	+	+	−	+	−	+	−
pattern 2	types of stimuli	Tur	Per	Air	Tur	Per	Air	Per	Tur	Air	Per	Tur	Air
	presence/absence of advance notice	−	+	−	+	−	+	−	+	−	+	−	+
pattern 3	types of stimuli	Air	Per	Tur	Air	Per	Tur	Per	Air	Tur	Air	Tur	Per
	presence/absence of advance notice	+	−	+	−	+	−	+	−	+	+	−	−

Per: percussion on the right side of the mandibular first premolar; Tur: no-load running of an air turbine beside the right side of the mandibular first molar with water spray for 10 s; Air: blowing air to the right side of the mandibular first molar for 5 s, the interval between each stimulus was 30 s.

**Table 2 ijerph-19-02540-t002:** The mean values of SN and PN for each item.

Explanatory Variable		N	SN		PN	
			Mean	SD	Mean	SD
sex	male	263	0.14 ^a^	0.07	−0.17 ^b^	0.04
	female	405	−0.09 ^a^	0.45	0.11 ^b^	0.06
age	20–29 years	120	−0.29	0.04	0.62	0.13
	30–39 years	159	−0.21	0.05	0.19	0.08
	40–49 years	64	−0.63	0.74	0.10	0.12
	50–59 years	76	0.51	0.13	−0.30	0.09
	60–69 years	161	0.25	0.11	−0.27	0.46
	70–79 years	88	−0.06	0.13	−0.50	0.47
type of stimulus	Air	228	−0.02	0.07	0.01	0.07
	Per	218	−0.05	0.07	−0.10	0.05
	Tur	222	0.07	0.59	0.09	0.71
advance notice	absent	335	0.04	0.06	0.02	0.06
	present	333	−0.04	0.04	−0.02	0.05
pattern of stimulation	1	236	−0.01	0.06	−0.13 ^c^	0.08
	2	237	0.05	0.07	−0.03 ^d^	0.06
	3	195	−0.05	0.07	0.19 ^c,d^	0.07
order of stimuli	1	57	−0.04	0.09	−0.10	0.08
	2	51	−0.01	0.10	−0.06	0.10
	3	55	−0.02	0.14	0.01	0.14
	4	58	−0.11	0.09	0.13	0.18
	5	57	−0.07	0.11	0.14	0.19
	6	55	0.01	0.11	0.17	0.19
	7	54	−0.12	0.08	−0.14	0.08
	8	58	−0.05	0.10	−0.10	0.08
	9	57	0.13	0.21	−0.06	0.12
	10	55	0.19	0.18	−0.08	0.09
	11	57	−0.03	0.09	0.09	0.13
	12	54	0.13	0.21	−0.00	0.14

^a–d^: The values with the same letters are significantly different (*p* < 0.05).

**Table 3 ijerph-19-02540-t003:** The results of the analysis with SN including all the factors simultaneously.

	Explanatory Variable	Estimated Value	Standard Error	95% Confidence Interval		*p*-Value
				Min	Max	
sex	male	0.05	0.37	−0.69	0.79	0.89
	female ^♦^	-	-	-	-	-
age		0.03	0.01	0.01	0.05	<0.01 **
types of stimuli	Air	−0.18	0.13	−0.44	0.08	0.17
	Per	−0.48	0.15	−0.77	−0.19	<0.01 **
	Tur ^♦^	-	-	-	-	-
advance notice	absent	0.06	0.25	−0.43	0.54	0.82
	present (reference)	-	-	-	-	-
pattern of stimulation	1	−0.76	0.47	0.11	−1.69	0.18
	2	−0.18	0.47	0.71	−1.12	0.76
	3 ^♦^	-	-	-	-	-
order of stimuli	1	−0.76	0.43	−1.60	0.09	0.08
	2	−0.27	0.43	−1.13	0.58	0.53
	3	−0.46	0.43	−1.31	0.38	0.28
	4	−0.85	0.43	−1.70	0.00	0.05
	5	−0.58	0.42	−1.42	0.25	0.17
	6	−0.61	0.43	−1.46	0.23	0.15
	7	−0.65	0.41	−1.46	0.17	0.12
	8	−0.77	0.40	−1.56	0.02	0.06
	9	−0.24	0.38	−0.99	0.50	0.52
	10	−0.16	0.34	−0.84	0.51	0.63
	11	−0.66	0.28	−1.21	−0.10	0.02 *
	12 ^♦^	-	-	-	-	-
stress-coping trait	Pla	0.12	0.07	0.08	−0.02	0.25
	Con	−0.08	0.09	0.37	−0.26	0.10
	See	0.12	0.06	0.04	0.01	0.24
	Acc	0.08	0.06	0.18	−0.04	0.21
	Sel	−0.11	0.07	0.12	−0.25	0.03
	Esc	0.10	0.10	0.30	−0.09	0.29
	Dis	0.07	0.08	0.43	−0.10	0.23
	Pos	−0.07	0.06	0.26	−0.20	0.05
interaction of stress-coping trait × presence/absence of advance notice	Pla × absent	0.01	0.04	−0.06	0.08	0.75
	Pla × present ^♦^	-	-	-	-	-
	Con × absent	−0.02	0.05	−0.12	0.07	0.66
	Con × present ^♦^	-	-	-	-	-
	See × absent	0.04	0.03	−0.03	0.10	0.26
	See × present ^♦^	-	-	-	-	-
	Acc × absent	0.01	0.03	−0.05	0.08	0.71
	Acc × present ^♦^	-	-	-	-	-
	Sel × absent	−0.01	0.04	−0.08	0.07	0.87
	Sel × present ^♦^	-	-	-	-	-
	Esc × absent	−0.05	0.05	−0.15	0.06	0.39
	Esc × present ^♦^	-	-	-	-	-
	Dis × absent	0.08	0.05	−0.01	0.17	0.09
	Dis × present ^♦^	-	-	-	-	-
	Pos × absent	−0.04	0.04	−0.11	0.03	0.28
	Pos × present ^♦^	-	-	-	-	-

**: *p* < 0.01, *: *p* < 0.05, ^♦^: reference.

**Table 4 ijerph-19-02540-t004:** The results of the analysis with PN including all the factors simultaneously.

	Explanatory Variable	Estimated Value	Standard Error	The 95% Confidence Interval		*p*-Value
				Min	Max	
sex	male	−8.1	12.6	−33.0	16.8	0.52
	female ^♦^	-	-	-	-	-
age		−1.5	0.3	−2.2	−0.8	<0.01 **
types of stimuli	Air	−9.5	4.9	−19.1	0.2	0.05
	Per	−14.5	5.4	−25.2	−3.9	<0.01 **
	Tur ^♦^	-	-	-	-	-
advance notice	absent	3.8	9.3	−14.5	22.1	0.68
	present ^♦^	-	-	-	-	-
the pattern of stimulation	1	−35.0	15.8	−66.3	−3.7	0.03 *
	2	−36.4	15.9	−67.9	−5.0	0.02 *
	3 ^♦^	-	-	-	-	-
order of stimuli	1	−15.3	15.2	−45.2	14.7	0.32
	2	−1.0	15.4	−31.2	29.2	0.95
	3	3.5	15.3	−26.5	33.5	0.82
	4	3.0	15.3	−27.1	33.1	0.85
	5	8.2	15.1	−21.4	37.7	0.59
	6	12.9	15.4	−17.3	43.1	0.40
	7	−14.1	14.8	−43.3	15.0	0.34
	8	−13.8	14.5	−42.3	14.8	0.34
	9	−11.3	13.7	−38.2	15.6	0.41
	10	−12.6	12.6	−37.3	12.1	0.32
	11	−6.7	10.5	−27.3	14.0	0.53
	12 ^♦^	-	-	-	-	-
stress-coping traits	Pla	−0.5	2.2	−4.9	4.0	0.84
	Con	−1.2	3.1	−7.3	4.8	0.69
	See	1.2	2.0	−2.8	5.2	0.55
	Acc	−0.6	2.1	−4.8	3.6	0.77
	Sel	−3.4	2.4	−8.1	1.2	0.15
	Esc	4.0	3.3	−2.5	10.5	0.22
	Dis	1.5	2.8	−4.1	7.1	0.59
	Pos	3.1	2.2	−1.2	7.4	0.16
interaction of stress-coping trait and presence/absence of advance notice	Pla × absent	0.5	1.4	−2.2	3.3	0.71
	Pla × present ^♦^	-	-	-	-	-
	Con × absent	−0.4	1.8	−4.0	3.1	0.81
	Con × present ^♦^	-	-	-	-	-
	See × absent	0.0	1.2	−2.4	2.4	0.99
	See × present ^♦^	-	-	-	-	-
	Acc × absent	1.6	1.3	−1.0	4.1	0.23
	Acc × present ^♦^	-	-	-	-	-
	Sel × absent	0.4	1.4	−2.4	3.2	0.80
	Sel × present ^♦^	-	-	-	-	-
	Esc × absent	1.4	2.0	−2.6	5.3	0.50
	Esc × present ^♦^	-	-	-	-	-
	Dis × absent	0.1	1.7	−3.3	3.5	0.97
	Dis × present ^♦^	-	-	-	-	-
	Pos × absent	−2.6	1.3	−5.2	0.0	0.04 *
	Pos × present ^♦^	-	-	-	-	-

**: *p* < 0.01, *: *p* < 0.05, ^♦^: reference.

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
