# Peer review of "The Effects of Providing Advance Notice and Stress-Coping Traits on Physiological Stress of Patients during Dental Treatment"

_ijerph, 2022, doi:10.3390/ijerph19052540_

Round 1

Reviewer 1 Report

The manuscript demonstrates interesting findings and considers relevant literature  

Below are few points for authors to consider:

1- Authors should specify the age of participants

2- Brief description of Lazarus Type Stress Coping Inventory is recommended to be added within the article 

3- What are the implication of the finding of this  study in the clinical setting  and what do authors propose in order to implement this finding in the clinic? 

Reviewer 2 Report

This article aims to use physiological measures of sympathetic and parasympathetic activation to measure the stress of simulated dental procedures and test for variations in the stress depending on reported coping style and whether or not advance warning is given of the procedure. This is an interesting idea that could add to what we understand about dental anxiety and fear. However there are many issues about how the study has been conducted and reported that weaken its claims and cast doubt on the conclusions of the study.

Introduction

The introduction starts with a statement that stress during dental treatment causes uneasiness and fear which may disturb routine dental checkups. This is an inaccurate description as in this context stress is more likely to be a response to treatment alongside uneasiness and fear. The references used to back this up are prevalence studies from 1992 and 1993. More recent references are available.

The statement that ‘Generally a person feels anxiety under unpredictable situations’ is an overgeneralisation and is supported by a reference from 1984 that refers to social phobia that is not relevant to the statement made.

On the basis of the above two observations please check all of your references for accuracy and recency as a large proportion of the reference list is from the 1980’s and 1990’s.

The introduction is short. It would be useful to have additional background to explain why providing advance notice of dental procedures might be considered as important in possibly influencing stress response and what work has been done looking at ways of coping with stress that support the further exploration of stress-coping traits in this context.

Materials and Methods

There is not enough detail in this section. There should be details of ethical review and approval of the study and of details of recruitment, for example how was this conducted, who was targeted, who approached them and how. Details of the monitor that was used to measure physiological stress would also be best placed here.

The use of physiological measures of sympathetic activity and parasympathetic activity would have been easier to understand if they had also been contextualised by subjective ratings. As far as I can read we have no way of knowing whether the subjects in this study had any form of dental anxiety or fear. Without this information ways of coping also appear to be irrelevant. It is possible that some of the positive results could have occurred by chance as the result of carrying out a large number of comparisons which do not appear to have been clearly derived from a theoretical rationale.

Results

Stress coping traits should be written in full for ease of reading.

Discussion

There is little evidence that the discussion considers and places the findings into context.

References

As I have said before, many of the references are from more than 20 years ago and there are more recent references available. There are many errors and incomplete references in the reference list.

Reviewer 3 Report

Thank you for your submission. The effects of providing advance notice and stress-coping traits on physiological stress of patients during dental treatment is an interesting topic, that requires further investigation. Please, consider the following points to improve your manuscript:

Abstract:

The sentence: “Subjects were given the simulated dental treatment including three stimuli, air, percussion and running of an air turbine, with or without advance notice of the stimulation during dental treatment.” is confusing. If the trial was performed by means of a simulated dental treatment, the sentence cannot conclude: with or without notice of the stimulation during dental treatment. Probably it could be replaced by: with or without notice of the stimulation during the simulation.

The same problem was stated with the sentence: “Real-time sympathetic nerve activity (SN) and parasympathetic nerve activity (PN) during the treatment were measured”. Please modify this sentence.

Materials and methods:

Was this trial examined by an Ethical Board?. Please, clarify it and include the acceptance code in the text.

It seems to be desirable to perform a previous dental examination before the stimulation. Was it done? Please, clarify it.

Discussion:

As stated in the previous comments, the authors performed a simulation. Then the sentence “Under the limitations of this study, we showed that the effects of advance notice of treatment just before treatment on physiological stress would depend on the stress coping traits of individual patients.” should be replaced by: “Under the limitations of this study, we showed that the effects of advance notice of treatment just before treatment simulation on physio-logical stress would depend on the stress coping traits of individual patients.”

The authors discuss in the Limitations subsection of the Discussion Section about the difficult extrapolation of the present results to the Japanese general population and recognize than Furthermore, “the stressors used in this study were simulated and non-invasive.”. However, the application of the present results in a real treatment environment depends on multiple factors, including the type of treatment and of course, the application on local anesthesia.

This topic: “the use of local anestesia” has a substantial contribution in vital signs during dental treatment, and it has only been cited in the sentence “SN has been reported to decrease in patients older than 40 years of age and increase in patients younger than 40 years of age when local anesthesia is used before tooth extraction [22].” in the discussion section.

This reviewer considers that the major limitation to apply the present results to a real treatment environment is the application of local anesthesia. It should be included in the limitations subsection and some reference should be cited.

References:

References 18 and 26 remain incomplete.

Round 2

Reviewer 2 Report

The authors have satisfactorily addressed the issues previously raised.